# Solving the SSVEP Paradigm Using the Nonlinear Canonical Correlation Analysis Approach

**DOI:** 10.3390/s21165308

**Published:** 2021-08-06

**Authors:** Danni Rodrigo De la Cruz-Guevara, Wilfredo Alfonso-Morales, Eduardo Caicedo-Bravo

**Affiliations:** 1Department of Electrical, Electronics and Telecommunications Engineering, Universidad de las Fuerzas Armadas, Sangolqui 171103, Ecuador; 2School of Electrical and Electronics Engineering, Faculty of Engineering, Universidad del Valle, Calle 13 #100-00, Cali 760032, Colombia; wilfredo.alfonso@correounivalle.edu.co (W.A.-M.); eduardo.caicedo@correounivalle.edu.co (E.C.-B.)

**Keywords:** information transfer rate, canonical correlation analysis, nonlinear canonical correlation analysis, steady-state visual evoked potentials, deep learning

## Abstract

This paper presents the implementation of nonlinear canonical correlation analysis (NLCCA) approach to detect steady-state visual evoked potentials (SSVEP) quickly. The need for the fast recognition of proper stimulus to help end an SSVEP task in a BCI system is justified due to the flickering external stimulus exposure that causes users to start to feel fatigued. Measuring the accuracy and exposure time can be carried out through the information transfer rate—ITR, which is defined as a relationship between the precision, the number of stimuli, and the required time to obtain a result. NLCCA performance was evaluated by comparing it with two other approaches—the well-known canonical correlation analysis (CCA) and the least absolute reduction and selection operator (LASSO), both commonly used to solve the SSVEP paradigm. First, the best average ITR value was found from a dataset comprising ten healthy users with an average age of 28, where an exposure time of one second was obtained. In addition, the time sliding window responses were observed immediately after and around 200 ms after the flickering exposure to obtain the phase effects through the coefficient of variation (CV), where NLCCA obtained the lowest value. Finally, in order to obtain statistical significance to demonstrate that all approaches differ, the accuracy and ITR from the time sliding window responses was compared using a statistical analysis of variance per approach to identify differences between them using Tukey’s test.

## 1. Introduction

Electroencephalography (EEG) signals are nonlinear, i.e., they present unstable characteristics and often vary in quality between user-to-user trials or even trials of the same user, posing significant challenges to build brain–computer interfaces (BCIs). The human brain is a complex nonlinear system, where the biosignals among these brain activities are ‘3N’: nonstationary, nonlinear, and noisy [1]; although, at small time windows of less than 4 s, the signal can be considered quasi-stationary [2], and the registering or discarding of these 3N characteristics can significantly influence the results. This is why a better analysis of EEG signals for steady-state visual evoked potentials (SSVEP) requires unconventional analyses and methods to address the nonlinear and nonstationary nature of these [3]. Additionally, a better analysis is also justified as a blinking visual stimulus and becomes weaker in EEG signals as the frequency of the stimuli increases [4], which results in a range of limited frequencies.

For almost a century, EEG signal acquisitions have demanded strict experimental conditions that include rigorous experimental protocols [5]. Improvements through the processing of signals, data analysis, and statistical modeling of the collected information from multiple electrodes to remove artifacts in time, frequency, or phase domains by proposing filters or pipelines as PREP [6], HAPPE [7], Automagic [8], or MADE [9] are being studied. However, for the SSVEP paradigm, the artifact-removal techniques are considered unnecessary due to the short period required to collect a trial.

The SSVEP paradigm is a natural response to visual stimuli with specific frequencies. The brain generates a response with the same spectrum coming from the retina stimulated by a flickering exogenous input [10]. The response generally occurs in the occipital and parietal lobes of the brain, where it is possible to collect multiple trials quickly. These stimuli produce steady-stable and small-amplitude visual evoked potentials if the eyes face the same exogenous input [11,12] and its frequency is greater than 2 Hz; otherwise, it produces a transient VEP. However, the stimulus achieves a periodic response called SSVEP when the frequency is higher (>6 Hz) [13]. The VEP responses also undergo an early transitional stage before entering SSVEP (see Figure 1, figure adapted from [14]). The VEP response initially has three components or waves: the P100 is a positive wave that appears 100 ms after the stimulus and is rounded by the other two minor, harmful waves N75 and N135, which appear 75-ms and 135-ms later than the stimulus, respectively. N135’s waveform depends on the contour process and changes in contrast when binocular interactions occur [15].

BCI systems based on SSVEP will achieve a better performance only when the responses overcome the transitional stage. Taking Figure 1, a displacement longer than 195 ms after the stimulus (P195) was observed as an effective starting point since the stability of the response for feature extraction is guaranteed. The time-frequency decomposition analysis showed a stable SSVEP result that started 250 ms after stimulus presentation at each fundamental frequency and its harmonic components [16]. Another relevant task is defining the minimal time windows to be analyzed. The time window is chosen in the first instance by the system developer and represents a trade-off between accuracy and classification speed [17]. A frequency greater than 6 Hz is necessary to define a steady-state into VEP responses. Therefore, the studies should collect at least three wavelengths to analyze the EEG signals more effectively [18], e.g., a 6 Hz stimulus requires a time window of 500 ms.

On the other hand, the amplitude and phase of SSVEP responses are highly sensitive to stimulus parameters such as repetition rate, color, luminance or modulation depth, and spatial frequency [19,20]. However, the SSVEP paradigm stands out for its minimal training capacity; robustness, high signal-to-noise ratio (SNR); and high information transfer rate (ITR) [21,22]. Furthermore, for the analysis and design of BCI systems, it is necessary to consider three fundamental aspects: the short processing time, the high information transfer rate (ITR), and the fact that these should be as noninvasive as possible [23,24]. SSVEP complies with all of these aspects since its records are minimally invasive, collects multiple trials in a short period, and provides a high information transfer rate [25]. A well-designed SSVEP-based BCI system would decrease ocular fatigue by reducing the user’s exposure to the flickering exogenous inputs.

Following the above idea, many researchers have developed different feature extraction methods for the SSVEP paradigm, mainly focused on the well-known CCA approach, since this has been proven to be efficient, stable, and simple to apply [26]. The most current review about BCI mainly shows CCA-based methods and others for feature extraction [27] such as the filter bank CCA, the multiset CCA, the individual template CCA, or the multilayer correlation maximization. The study also includes other linear approaches such as task-related component analysis (TRCA), correlated component analysis (CORCA), and least absolute shrinkage and selection operator (LASSO). However, as we mentioned before, the EEG signals are nonlinear; so, approaches such as empirical and variational mode decomposition (i.e., EMD and VMD as nonlinear) have had outstanding performances, as mentioned in Labecki et al. [28], De la Cruz et al. [29].

Based on the above statement, this paper presents the nonlinear CCA (NLCCA) implementation to classify EEG signals using the SSVEP paradigm. No reference to NLCCA as a solution for SSVEP-based BCI systems was established, although the method has been used since its beginnings in other applications, especially climatic ones, such as the work of [30], and in more current ones such as [31]. Besides, we also compared its performance with other approaches such as CCA and LASSO. The first method represents the basis to show the treatment effects of the signals; the second is a simple approach that has served to obtain better performance than CCA [32]. Thus, this paper presents the average accuracy and ITR as comparison metrics using different time windows in two scenarios: full-users and removing the worst user. First, the best time window base on the average ITR value was defined. Next, an inspection of the phase effects during the time sliding window was carried out by taking into account and discarding the transitional stage issue, i.e., using EEG data when the stimuli appeared without shifting and 200 ms-later; here, the coefficient of variation metric was used. Finally, a statistical analysis was conducted to identify differences between the approaches using analysis of variance (ANOVA) and post-Tukey’s test [33,34].

## 2. Materials and Methods

In this study, an externally provided database was used. This dataset corresponds to the experiments developed at the Swartz Center for Computational Neuroscience, Institute for Neural Computation, from the University of San Diego [35] (Dataset available in https://www.kaggle.com/lzyuuu/ssvep-sandiego (accessed on 7 June 2021)). The dataset contains the records of ten volunteers without any reported illnesses who observed twelve visual stimuli while an EEG system captured their signals. The following subsections present a description of the main characteristics of the dataset that are relevant to the study (Section 2.1, Section 2.2 and Section 2.3), the implemented approaches (Section 2.4), and the used performance metrics (Section 2.5).

### 2.1. Experimental Setup and Volunteers

Ten healthy, right-handed volunteers (S1–S10) with standard or normal-corrected vision participated in the experiments (nine men and one woman with an average age of 28 years). They sat in a comfortable chair positioned 60 cm in front of a 27 in LCD monitor (60 Hz refresh rate, 1280×800 screen resolution) in a low-illuminated room.

### 2.2. Experimental Protocol

Each volunteer completed 15 blocks, and each block consisted of 12 trials corresponding to each stimulus (from 9.25 Hz to 14.75 Hz at 0.5 Hz intervals). In each trial, the user had one second to fix his/her gaze to the target stimulus position, which was indicated by a red square marker. Subsequently, all stimuli started to flicker simultaneously for 4 s on the monitor. It is important to note that the marker was randomly located for each new trial until the volunteer ended a block [35]. Figure 2 shows the time of each trial.

### 2.3. EEG Signal Recordings and Preprocessing

The sampling frequency of EEG signal recordings was found to be 256 Hz. Eight Ag/AgCl electrodes covered the occipital area using a BioSemi ActiveTwo EEG system (Biosemi, Inc. Amsterdam, The Netherlands), as shown in Figure 3, corresponding to electrodes O1, O2, Oz, PO7, PO3, POz, PO4, and PO8.

The raw recordings were then filtered using a bandpass between 6 Hz and 80 Hz, with an infinite impulse response (IIR) filter. Here, zero-phase forward and reverse IIR filtering was implemented as suggested in [35].

### 2.4. SSVEP Pattern Recognition Approaches

#### 2.4.1. Least Absolute Shrinkage and Selection Operator (LASSO)

LASSO is a scattered regression model that offers high interpretable regression coefficients. Zhang et al. [36] proposed the implementation of the LASSO method for the detection of SSVEP signals [36], and it did in fact prove to be useful and robust in the feature extraction and selection of EEG signals into the SSVEP paradigm. LASSO’s main purpose is to solve a standard linear regression model [37,38].

LASSO obtains an approximation of the $g_{n}$ term in the SSVEP model denoted in
(1)$$y_{n}=X_{f}g_{n}+e_{n}$$
where $y_{n}$ represents the EEG signal, $X_{f}$ the SSVEP reference signal, and $e_{n}$ is an additive noise vector [39]. $\hat{\beta }$ is defined as the LASSO estimator and is calculated using quadratic programming, as given below [40]:(2)$$\hat{\beta }=\underset{\beta }{arg\min}\left({∥y-X\beta ∥}_{2}^{2}+\lambda {∥\beta ∥}_{1}\right),$$
where ${∥\cdot ∥}_{1}$, ${∥\cdot ∥}_{2}$ denote the norm $l_{1}$ and the norm $l_{2}$, respectively; λ is a penalty parameter that provides a sparse solution (i.e., it forces many entries to zero). Thus, the optimization problem is to find the optimal $\hat{\beta }$ vector—here, $\hat{\beta }={\left[\beta_{1,1},\ldots ,\beta_{1,H},\ldots ,\beta_{F,1},\ldots ,\beta_{F,H}\right]}^{T}$, where *H* is the representation of the number of harmonics. Each element indicates the level of contribution of the stimulus frequency *f* to the EEG signal. The largest LASSO estimator is established by the target stimulus, which is the one that provides the highest contribution and is defined as
(3)$$f_{t}=\underset{f}{max}\frac{\sum_{i=1}^{N}\sum_{h=1}^{H}\left|\beta_{f,h}^{i}\right|}{N},\;\;\;f\in \left\{f_{1},\ldots ,f_{F}\right\},$$
where *N* is the number of channels of the EEG signal. LASSO is considered a subject-independent technique since the penalty parameter that influences the LASSO performance is calculated offline by taking advantage of the SSVEP data from multiple subjects [41]. There has been recent research where the LASSO method was applied because of the ability to perform multichannel analysis for identifying SSVEP signal frequencies (e.g., [32,42]).

#### 2.4.2. Canonical Correlation Analysis (CCA)

CCA is a feature extraction method that is widely used to detect the frequency of EEG signals through the underlying correlation between two multidimensional variables. The idea of CCA for detection of the SSVEP paradigm is to take the EEG signal and relate it to a set of frequencies that coincide with the frequencies of the target stimuli and establish the highest multidimensional correlation [43].

The first multidimensional variable (*x*) indicates the multichannel EEG signals. The second (*y*) refers to sinusoidal reference signals, which correspond to the primary and other harmonics (usually the second and third) from the fundamental frequency of the observed stimulus [44]. This is given by
(4)$$y=\left(\begin{array}{c} \sin(2\pi f_{k}t)\\ \cos(2\pi f_{k}t)\\ \vdots \\ \sin(2\pi N_{h}f_{k}t)\\ \cos(2\pi N_{h}f_{k}t) \end{array}\right);\;\;\;t=\frac{1}{S},\frac{2}{S}\ldots \ldots \frac{T}{S},$$
where $f_{k}$ is the stimulus frequency, $N_{h}$ is the number of harmonics, *T* is the number of sampling points, and *S* is the sampling rate.

Then, a pair of linear combinations $u=x^{T}W_{x}$ and $v=y^{T}W_{y}$, called canonical variates, are calculated by tuning the weight vectors $W_{x}$ and $W_{y}$ in order to ensure the correlation between them achieves a maximum value. Notice that each stimulus will have its weight vectors for frequency, so there is a need to calculate the canonical variates from the multidimensional variables (i.e., EEG and reference signals). Subsequently, the correlations between canonical variates (CCA coefficients) were calculated. The maximum coefficient argument of these canonical variates per frequency should correspond to the stimulus [45]. A summary of the above explanation is given by
(5)$$\hat{k}=\underset{k}{arg\max}\left(\rho_{k}\right),\;\;\;\;k=1,2,\ldots ,K,$$
where ρ are the CCA coefficients, *K* is the stimulus frequency number, and $\hat{k}\in \left[1,\;K\right]$ is the selected stimulus.

#### 2.4.3. Nonlinear Canonical Correlation Analysis (NLCCA)

The NLCCA method follows the same approach as the CCA base technique. The modification is found in the linear mappings of Equation (Equation 6), by nonlinear mapping functions with the use of neural networks (NNs). In Figure 4 found on the left, the mappings from *x* to *u* and *y* to *v* are represented by the double-barreled NNs [46].

Consider one dataset $x_{i}\left(t\right)$ with variables *i* and another dataset $y_{j}\left(t\right)$ with *j* variables, where each dataset has t=1,…,n samples. Variables $x_{i}\left(t\right)$ can be grouped together to form vector x(t) and variables $y_{j}\left(t\right)$ can be grouped together to form vector y(t). As shown previously, CCA obtains the linear combinations according to
(6)$$u\left(t\right)=x{\left(t\right)}^{T}\cdot W_{x},\;\;\;v\left(t\right)=y{\left(t\right)}^{T}\cdot W_{y}$$
where $W_{x}$ and $W_{y}$ are weight vectors that represent correlated spatial patterns corresponding to the fields of the EEG data. However, the NLCCA corresponds to *u* and *v*, and $u=f(\omega_{x},x)$ and $v=f(\omega_{y},y)$ are functions where $\omega_{x}$ and $\omega_{y}$ are all neural network parameters that require establishing. Therefore, a learning procedure such as the CCA approach is required to maximize the correlation between *u* and *v*.

Based on Figure 4, the networks on the left assign nonlinearly $x\overset{}{\to }u$, $y\overset{}{\to }v$ depending on
(7)$$\begin{array}{cc} h^{\left(x\right)}=\tanh\left[x^{T}W^{\left(x\right)}+b^{{\left(x\right)}^{T}}\right],\; & u={\tilde{w}}^{\left(x\right)}\cdot h^{{\left(x\right)}^{T}}+{\tilde{b}}^{{\left(x\right)}^{T}}\\ h^{\left(y\right)}=\tanh\left[y^{T}W^{\left(y\right)}+b^{{\left(y\right)}^{T}}\right],\; & v={\tilde{w}}^{\left(y\right)}\cdot h^{{\left(y\right)}^{T}}+{\tilde{b}}^{{\left(y\right)}^{T}} \end{array}$$
where $h^{\left(x\right)}$ and $h^{\left(y\right)}$ are the output nodes of the hidden layer and tanh[·] is the hyperbolic tangent function. The neural network parameters are $W^{\left(x\right)}$ and $W^{\left(y\right)}$ as the hidden layer weight matrices, $b^{\left(x\right)}$ and b(y) as the hidden layer bias vectors, ${\tilde{w}}^{\left(x\right)}$ and ${\tilde{w}}^{\left(y\right)}$ as the weight vectors of the output layer, and ${\tilde{b}}^{\left(x\right)}$ and ${\tilde{b}}^{\left(y\right)}$ as the bias vectors of the output layer.

Next, the Pearson correlation in the first stage of the architecture, $\mathrm{cor}\left(u,v\right)$, between the canonical variables *u* and *v* should be maximized. The Pearson correlation is given by
(8)$$r=\mathrm{cor}\left(u,v\right)=\frac{n\sum uv-\sum u\sum v}{\sqrt{n\sum u^{2}-{\left(\sum u\right)}^{2}}\sqrt{n\sum v^{2}-{\left(\sum v\right)}^{2}}}$$
where *n* is the length of any vector *u* or *v*.

The neural network parameters were obtained by minimizing the cost function
(9)$$C_{1}=-\mathrm{cor}\left(u,v\right)+{\langle u\rangle }^{2}+{\langle v\rangle }^{2}+{\left({\langle u^{2}\rangle }^{\frac{1}{2}}-1\right)}^{2}\;+{\left({\langle v^{2}\rangle }^{\frac{1}{2}}-1\right)}^{2}+P_{1}[\sum_{ki}{\left(W_{ki}^{\left(x\right)}\right)}^{2}+\sum_{lj}{\left(W_{lj}^{\left(y\right)}\right)}^{2}]$$

In the above expression, $\langle g\rangle$ denotes the sample or temporal mean, i.e., $\frac{\sum q}{n}$. The first term maximizes the correlation between the canonical variables *u* and *v*; the second and third terms represent the sum of the squared samples; the fourth and fifth terms are normalization constraints that force *u* and *v* to have a zero mean and unit variance. The sixth term is a regularization term ($L_{2}$-norm type) to avoid overfitting in the neural networks, and its relative magnitude is controlled by $P_{1}$. Larger $P_{1}$ values lead to smaller weights (fewer effective model parameters), resulting in a more linear model [46].

The right-hand side of the two neural networks, shown in Figure 4, represents the inverse mappings for $\hat{x}$ and $\hat{y}$ with the computed canonical variables *u* and *v*.
(10)$$h_{k}^{\left(u\right)}=\tanh\left[{\left(w^{\left(u\right)}u+b^{\left(u\right)}\right)}_{k}\right],\;\hat{x}={\tilde{W}}^{\left(u\right)}h^{\left(u\right)}+{\tilde{b}}^{\left(u\right)}\;h_{l}^{\left(v\right)}=\tanh\left[{\left(w^{\left(v\right)}v+b^{\left(v\right)}\right)}_{l}\right],\;\hat{y}={\tilde{W}}^{\left(v\right)}h^{\left(v\right)}+{\tilde{b}}^{\left(v\right)}$$

To find the bias and weight parameters of the two networks on the right, the cost functions were minimized, as given by
(11)$$C_{2}=\langle {∥\hat{x}-x∥}_{2}^{2}\rangle +P_{2}\sum_{k}{\left(w_{k}^{\left(u\right)}\right)}^{2}\;C_{3}=\langle {∥\hat{y}-y∥}_{2}^{2}\rangle +P_{3}\sum_{l}{\left(w_{l}^{\left(v\right)}\right)}^{2}$$
where ${∥\cdot ∥}_{2}^{2}$ is the square of $L_{2}$-norm, with $L_{p}$-norm as in
(12)$$L_{p}\left(e\right)={\left({∥e∥}_{p}\right)}^{1/p}={\left(\sum_{i}{\left|e_{i}\right|}^{p}\right)}^{1/p}$$

The mean square error (MSE) is given by the first term of $C_{2}$ and $C_{3}$ in Equation (Equation 11). $P_{2}$ and $P_{3}$ correspond to penalty terms of the synaptic weights for the neural network.

However, in this study, the robust version of NLCCA was used, as presented by Cannon and Hsieh in 2008, since the previous architecture presented by Hsieh in 2001 is sensitive to overfitting. This modified robust version involved two changes. First, the section that calculates the MSE within the cost functions $C_{2}$ and $C_{3}$ are replaced by the $L_{1}$-norm in order to calculate the mean absolute error (MAE). Second, the biweight midcorrelation (bicor) is used instead of the Pearson correlation on the $C_{1}$ cost function to measure the similarity between the canonical variables *u* and *v*, i.e., the $\mathrm{cor}\left(u,v\right)$ is replaced by $\mathrm{bicor}\left(u,v\right)$.

The literature has shown the advantages of nonlinear CCA methods over linear CCA methods in several applications (e.g., [47,48]). However, the nonlinear methods have not been explored much, and therefore have few recent studies related to, most notably, BCI systems (e.g., [49]), where the application areas are primarily found in forecasting and climate studies [50]. In this study, the 2008 version mentioned above was used to apply NLCCA in the feature extraction of EEG–SSVEP signals.

The procedure for calculating the biweight midcorrelation and Pearson’s correlation coefficient is similar, with the exception that robust parameters replace the nonrobust parameters (i.e., covariance, mean, and expected deviation). The bicor function forecasts *u* from *v* and vice-versa, as it was calculated in the standard NLCCA model presented in Equation (Equation 17).

The biweight midcorrelation function for two vectors *x* and *y* ($\mathrm{bicor}\left(x,y\right)$) is calculated, first defined as the median of the vector *x* (med(x)) and the median absolute deviation (mad(x)); then, $u_{i}$ and $v_{i}$ are determined as [51], where
(13)$$u_{i}=\frac{x_{i}-med\left(x\right)}{9\;mad(x)},\;v_{i}=\frac{y_{i}-med\left(y\right)}{9\;mad(y)}$$
where *i* is the number of items for each vector *x* and *y*.

The weights $w_{i}^{\left(x\right)}$ and $w_{i}^{\left(y\right)}$ are defined as
(14)$$w_{i}^{\left(x\right)}={\left(1-u_{i}^{2}\right)}^{2}I\left(1-\left|u_{i}\right|\right)\;w_{i}^{\left(y\right)}={\left(1-v_{i}^{2}\right)}^{2}I\left(1-\left|v_{i}\right|\right)$$
where I represents the identity function, where
(15)$$I\left(x\right)=\left\{\begin{array}{cc} 1, & \mathrm{if}\;x>0\\ 0, & \mathrm{otherwise}. \end{array}\right.$$

Then, it is normalized to get a sum of the weights equal to one as follows:(16)$${\tilde{x}}_{i}=\frac{\left(x_{i}-med\left(x\right)\right)w_{i}^{\left(x\right)}}{\sqrt{\sum_{j=1}^{m}{\left[\left(x_{j}-med\left(x\right)\right)w_{j}^{\left(x\right)}\right]}^{2}}}\;{\tilde{y}}_{i}=\frac{\left(y_{i}-med\left(y\right)\right)w_{i}^{\left(y\right)}}{\sqrt{\sum_{j=1}^{m}{\left[\left(y_{j}-med\left(y\right)\right)w_{j}^{\left(y\right)}\right]}^{2}}}$$

The biweight midcorrelation function finally is determined as
(17)$$\delta =\mathrm{bicor}\left(x,y\right)=\sum_{i=1}^{m}{\tilde{x}}_{i}{\tilde{y}}_{i}$$

The use of bicor and MAE functions in the robust NLCCA model leads to a more stable algorithm, with improved performance and a decreasing sensitivity to overfitting [43]. These characteristics make robust NLCCA an excellent approach to analyze databases with low signal-to-noise ratios such as electroencephalographic (EEG) signals.

The cost function $C_{1}$ must be derived to implement the backpropagation algorithm; however, it is possible to use numerical approximation methods to achieve it. In order to obtain an effective tuning, the next stage is transitioned to by following an interactive process where the goal is to guarantee that all functions achieve minimum values. By inspection, $C_{1}$ should see a value near −1, and $C_{2}$ and $C_{3}$ values near 0 are expected. The derivative of $\mathrm{bicor}\left(x,y\right)$ is represented as follows:(18)$$\begin{array}{c} \frac{\partial }{\partial x_{i}}\left(\mathrm{bicor}\left(x_{i},y_{i}\right)\right)=\frac{{\tilde{y}}_{i}\left(1-{{\tilde{x}}_{i}}^{2}\right)\left[w_{i}^{\left(x\right)}-4u_{i}^{2}\left(1-u_{i}^{2}\right)I\left(1-\left|u_{i}\right|\right)\right]}{\sqrt{\sum_{j=1}^{m}{\left[\left(x_{j}-med\left(x\right)\right)w_{j}^{\left(x\right)}\right]}^{2}}} \end{array}$$

Depending on which path (*u* or *v*), the first part of the network involves taking, from which data is derived. Therefore, if the output *v* is constant, *u* is derived, and vice versa. Thus, the corresponding changes of the left-hand neural network parameters are taken to later verify that the cost function achieves the minimal values.

Subsequently, each stimulus passes through the trained neural networks to obtain both *u* and *v* vectors, where the bicor function is expected to obtain the highest correlation. From the nonlinear point of view, one advantage is that by maximizing the correlation, there is a more discriminating factor between those that belong to a specific stimulus and those that do not. In CCA, the linear representation that is relatively coarse brings with it differences between the incoming stimulus, resulting in little effective discrimination concerning those found in the templates.

For each template, a neural representation is necessary as it becomes the element of comparison concerning a new arrival of a stimulus, to then verify the $\mathrm{bicor}\left(u,v\right)$ through its transformation, (N.B. an approximation given by $\hat{u}$ is provided). All templates have a specific *v* that is to be compared with a new input signal that will generate a vector $\hat{u}$, where the highest correlation indicates the possible observed stimulus, as shown in the following expression:(19)$$\hat{k}=\underset{k}{arg\max}\left(\delta_{k}\right),\;\;\;\;k=1,2,\ldots ,K,$$
where $\delta_{k}$ is the correlation coefficient between $\hat{u}$ and $v_{k}$, as obtained from the NLCCA approach, with k=1,…,K trained templates.

### 2.5. Performance Metrics

The information transfer rate introduced by [52] as a performance measure for BCIs has been preferred by researchers. The measures for calculating the ITR are related to how the user performs the task quickly and effectively, the task speed time, the performance time, the possibility of completing the task, and the selected time being involved in efficiency measures.

For a BCI system with *N* targets and a general classification precision *P*, the ITR can be defined as the amount of information reported per unit of time, which depends on both the transmission speed and the precision. In BCIs, the bit rate per trial, *B* (bits/trials), is expressed as
(20)$$B=\log_{2}N+P\log_{2}P+\left(1-P\right)\log_{2}\frac{1-P}{N-1},$$
where the transfer rate *B* can be translated into bits/minute, once it is multiplied by the frequency of decisions per minute *T*, as permitted in the BCI system shown in expression (Equation 21).
(21)$$\mathrm{ITR}=\frac{B}{T}$$

The performance of the BCI system can be determined by calculating the precision (Acc—Accuracy), and is defined as
(22)$$Acc=\frac{TP+TN}{TP+TN+FP+FN}\times 100,$$
where TP and TN represent the total number of correctly detected true positive events and true negative events, respectively. FP and FN represent the total number of incorrect positive events and incorrect negative events, respectively [53].

## 3. Analysis of Results

This section assesses the potential of the NLCCA approach. First, the accuracy and ITR performance per user was verified by using different time windows in order to identify the most appropriate. Later, all users were grouped together, and the worst was removed (User 3) to show the average accuracy and ITR performance. This first analysis shows the general performance of each approach. The following analysis seeks to verify the time sliding window response using the best established time window. Following the flickering exposure, here, two scenarios immediately after and around 200 ms were presented in order show the transient response in VEPs. The stationary responses to measure each coefficient of variation were taken to define the robustness associated with phase shifting. Finally, an analysis of variance (ANOVA) with a post- Tukey’s test was performed, where the time sliding window for one second per approach was used to define the most suitable.

### 3.1. Accuracy and ITR Inspection per User

The recognition accuracy of each subject for 12 stimuli frequencies and the corresponding ITR derived from the three approaches is shown in Figure 5 and Figure 6. The red, blue, and green lines represent the CCA, Lasso, and NLCCA approaches.

The results show a clear tendency for 9 of 10 subjects to show a higher performance for the NLCCA method, followed by the CCA, and finally the LASSO method. User 3 shows inferior performance for all approaches, which is not consistent with the other results, suggesting that the user’s data contains possible errors. As a result, a decision was made to exclude this user from the analysis.

Figure 7a,b present the average accuracy and ITR for 10 subjects, while Figure 8a,b exclude user 3. These last two graphs show that the NLCCA method performs better than the other two, especially within the first 2 s; further, later, it converges with the CCA method. Based on these results, a good performance was evidenced to have been obtained with smaller time windows using NLCCA. In addition, Figure 8a,b show that in the second time window, the NLCCA approach achieves a recognition accuracy of 73%, which is considerably higher than the 48.4% and 25.6% of CCA and LASSO, respectively. Here, one second is determined to be sufficient for producing a suitable performance, since the difference between the ITR results for one second and one and a half seconds are undistinguished, as checked by their variances. Besides, a short time window to reduce the fatigue is part of the objective, so one second as a reference value was chosen.

Table 1 illustrates the maximum ITR value per user, and the corresponding recognition accuracy and time window. The last two rows show the average of all the results (excluding user 3). In general, Table 1 reflects a better performance for the NLCCA approach against the others, where NLCCA obtained the highest ITR values for 8 of the 9 users (1, 2, 4–7, 9, and 10)—user 8 being the exception with only 1.1 (bit/min) below the CCA. In addition, the time taken to obtain these highest ITR values was shorter than with the other approaches, with an average of 1.28 s for NLCCA. In contrast, the time taken to obtain the highest ITR values for the CCA and LASSO corresponded to 2.39 and 2.89 s, exceeding the NLCCA by more than one second. The LASSO method obtained the lowest performance in all cases.

### 3.2. Transient and Steady-State Responses by the Time Sliding Windows (Phase Effects)

The Accuracy and ITR data shown below were analyzed to determine any effect when the phase is modified. Here, a one-second sliding window in steps of 3.9 ms was taken (one for each sample), and its effects with and without transient response for the average performance was taken for all users except User 3. Figure 9 and Figure 10 show the results obtained by including the transient response, i.e., taking data immediately after the flickering exposure. Figure 11 and Figure 12 show the results around 200 ms after the flickering, when the steady-state response for the average performance with all users excluding User 3 was taken, respectively.

From Figure 9, Figure 10, Figure 11 and Figure 12, the phase-shifting was observed to have produced nocive effects. It can be observed that when using the full signal (Figure 9 and Figure 10), the three methods show a drop in performance during the first 500 ms when compared to the same performance after 500 ms. However, the most affected method is identified as the CCA, since the drop in its performance is considerably greater. However, when using the signal without the first 200 ms, i.e., the transient (Figure 11 and Figure 12), the initial performance drop in all three methods is completely eliminated. Here, the initial performance is seen to remain in the same range of variation as the rest of the signal.

When comparing this behavior to previous studies, it is important to highlight the need for achieving a steady-state in order to obtain the best performance in each approach. However, the study also revealed that LASSO and NLCCA still offer some resistance to the transient response, while CCA does not. Although both ITR results evidence no significant change, NLCCA is much more suitable when the shifting immediately after the flickering exposure is ignored.

On the other hand, considering the steady-state (around 200 ms later), the phase-shifting effects are still observed. These effects produce variations in the accuracy and ITR performances, which were analyzed using the coefficient of variation. In addition, some slight differences that appear between NLCCA with and without transient responses were observed, which other approaches do not have. These differences are due to different neural networks being trained for each procedure, although the data quality facilitates a better performance. Table 2 shows the mean, standard deviation, and coefficient of variation data for each of the signals presented in Figure 11 and Figure 12. It can be seen that the mean of NLCCA is higher than the other two methods, both in precision (80–86) and ITR (72–79), and the standard deviation is similar in all three methods. This results in the coefficient of variation of NLCCA being the lowest in all cases (2.37 to 4.22), thus indicating that this method is more robust.

### 3.3. Identifying the Best Approach

Table 3 shows the analysis of variance performed on the obtained values, as shown in Figure 11 and Figure 12. The calculated *F*-value for all cases (from 44,000 onwards) was observed and, concerning the reliability of 95%, this was found to consistently be more significant than the *F*-value of 3. This result indicates that there are significant differences between the three analyzed approaches.

Finally, to determine which approach these differences can be found in, and to identify the best performance, the post-Tukey’s test was used, and Table 4 shows the results. According to the table, all differences between each case exceed the DVS value, indicating significant differences between all the methods. Therefore, the NLCCA ($U_{1}$) is shown to have the best performance, followed by CCA ($U_{2}$), and finally LASSO ($U_{3}$). In addition, these results correspond to a one-second sliding window, where a suitable enough flickering exposure to avoid fatigue was defined.

This paper presents a first analysis of the NLCCA performance compared to the CCA and LASSO approaches. The NLCCA approach was found to be more effective than the other two, especially in the first 2 s, where it reached higher ITR values, thus implying that NLCCA can obtain an acceptable result in less time. On the other hand, due to the nonlinear nature of brain signals, nonlinear methods would respond more appropriately than linear methods, which is typical of EEG signals [28]. Therefore, this paper shows the implementation of the NLCCA approach and the CCA approach, the latter being nonlinear and the most widely used in this field.

The NLCCA, based on neural networks, is characterized by being adaptive and representative of the CCA model in a nonlinear mode [46]. Its complex structure and the use of nonrobust functions aid in the performing of an overfitting process, mainly when manipulating noisy and short databases. The robust NLCCA solved these issues by using robust cost functions, explicitly replacing the inverse mapping MSE network with MAE, and changing the Pearson correlation by the biweight midcorrelation in the double barrier network. The robust NLCCA is expected to be ideal for EEG signals due to the biosignals generally being nonstationary, nonlinear, and noisy [1]. Indeed, the implementation showed that NLCCA increases their stability when the data quality exclusively includes the steady-state responses, showing an outstanding performance when the database is limited in size.

## 4. Conclusions

The NLCCA approach for EEG signal treatment into the SSVEP paradigm shows its potential as it offers outstanding performance, as evaluated using statistical analysis. All its results showed a better performance in terms of accuracy and ITR. Here, the average behavior for the 12 stimuli per user was evaluated. Later, the average and standard deviation in both scenarios (with and without User 3) was calculated, where NLCCA obtained the shortest time windows in almost all subjects with the best ITR; the results also showed that a one-second window was enough to obtain suitable performance. It is crucial to reduce the time window with a high ITR since this reduces user fatigue during the experimental stage.

A second experiment determined the phase effects using the time sliding window; this experiment combined an additional interaction to include the transient response into the analysis. Several studies have reported that the steady-state occurs approximately after N135, while others indicated it occurred after 200 ms [54]. Figure 9 and Figure 10 showed an initial impairment for each one of the responses. The findings of this study showed that NLCCA tries to keep a low variability as LASSO did also, but these findings are incomparable due to the significant difference in performances. The CCA was also observed to not be significantly affected by the transient response. In addition, in being able to identify the time when the flickering exposure appears in a BCI development by taking the attendance user, the different methods will show a good performance. However, with a lack of attendance control, the NLCCA seems more appropriate based on ITR performance.

On the other hand, the NLCCA performance showed that the signals in steady-state slightly improved due to the neural networks being newly trained. LASSO and CCA approaches do not have this advantage since they are deterministic. Despite this, the variations due to the phase effects are still evident with all approaches. Nonetheless, the coefficients of variation to determine the best approach were verified, identifying the NLCCA to have the lowest values when compared to the others.

Finally, the third experiment involved the one-second sliding window data to design the analysis of variance to identify differences between the approaches. The scenarios showed differences between them, so a post-Tukey’s test to define the statistical differences was used. The results helped to state that the NLCCA approach presented the best performance.

Future studies could aim to investigate the behavior of the NLCCA approach when under other operating scenarios, always focused on the objective frequency discrimination stage. The analysis of new research studies will improve validation parameters such as ITR, accuracy, and reduced time window to mitigate user fatigue as much as possible. One way to explore these issues is to use preprocessing methods to improve the signal quality before implementing NLCCA as the empirical or the variational mode decomposition.

## Figures and Tables

**Figure 1 sensors-21-05308-f001:**
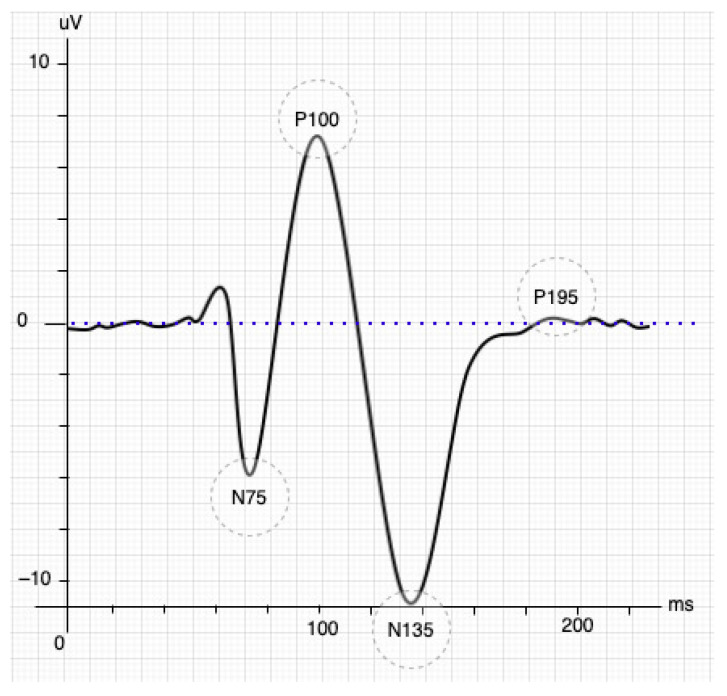
Stimulation of the inverted pattern of the SSVEP signal [14].

**Figure 2 sensors-21-05308-f002:**
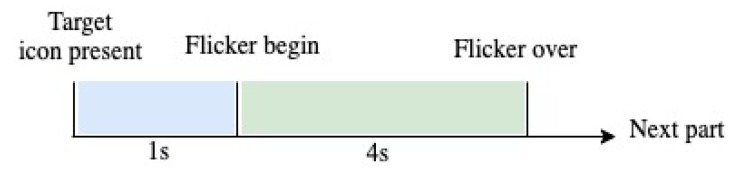
Distribution of the capture time of the SSVEP signal.

**Figure 3 sensors-21-05308-f003:**
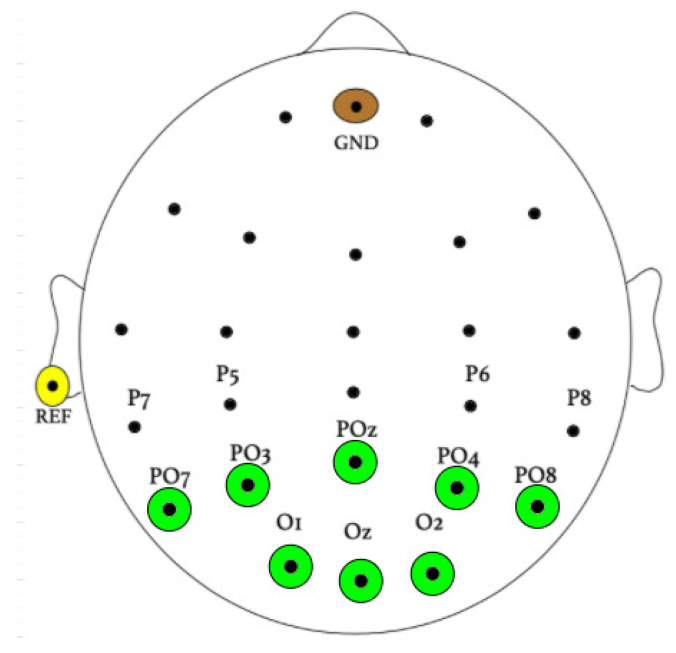
Electrode placement during experimentation.

**Figure 4 sensors-21-05308-f004:**
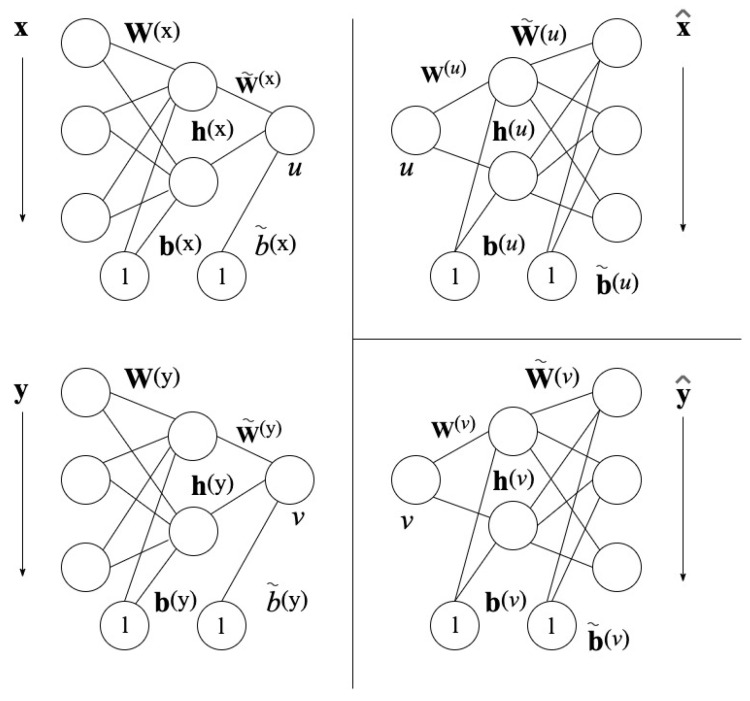
Neural Network Architecture for NLCCA [43].

**Figure 5 sensors-21-05308-f005:**
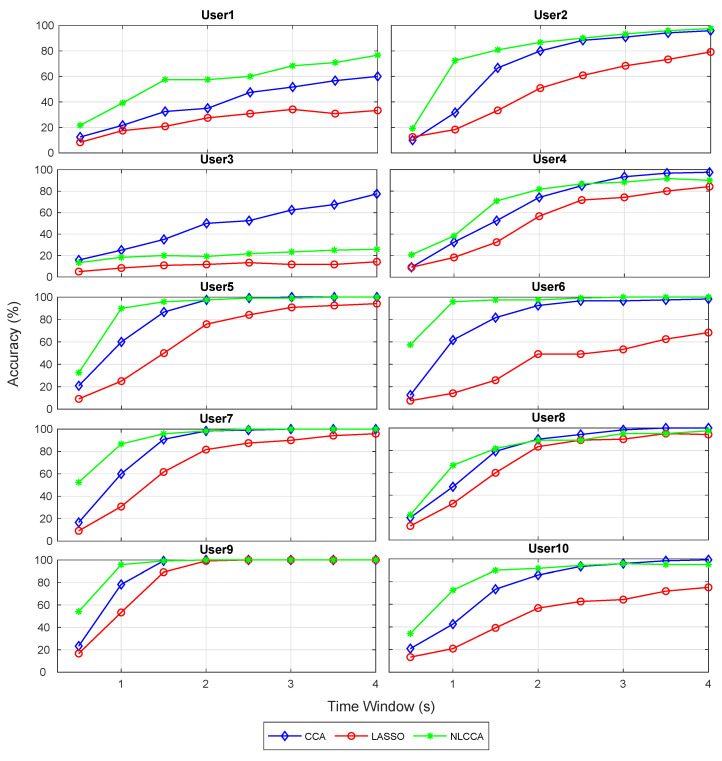
Comparison of accuracy for each subject between CCA, LASSO, and NLCCA.

**Figure 6 sensors-21-05308-f006:**
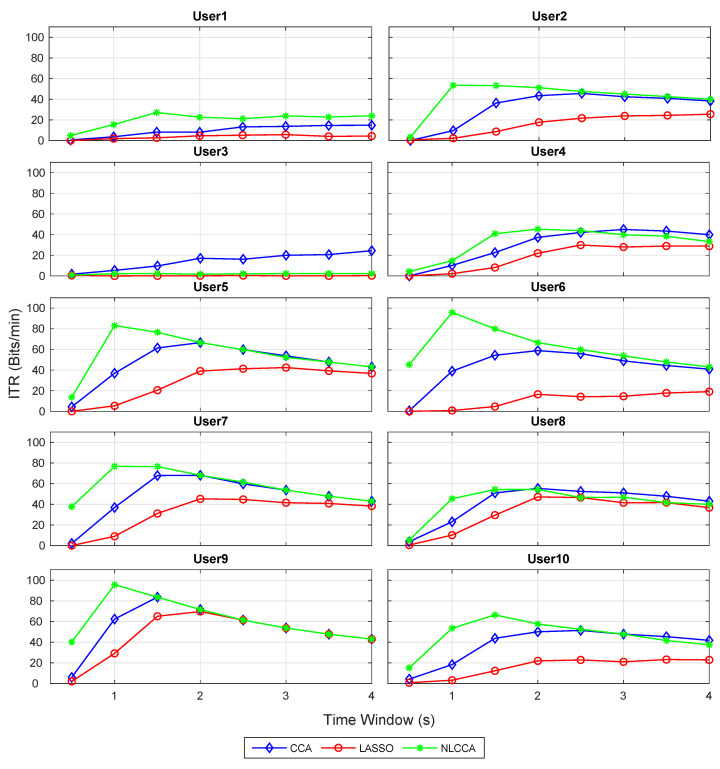
Comparison of ITR (information transfer rate) for each subject between CCA, LASSO, and NLCCA.

**Figure 7 sensors-21-05308-f007:**
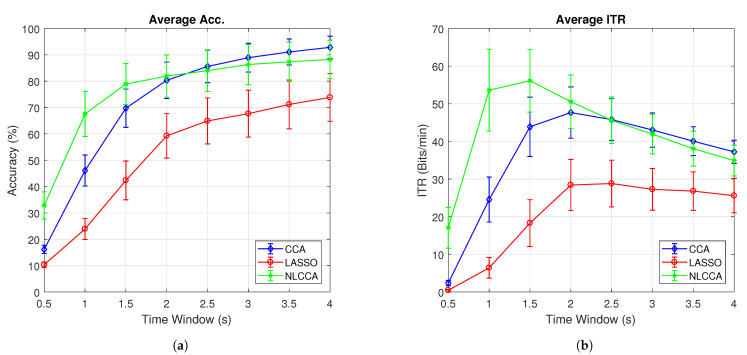
Comparison of (**a**) average accuracy and (**b**) average ITR between CCA, LASSO, and NLCCA.

**Figure 8 sensors-21-05308-f008:**
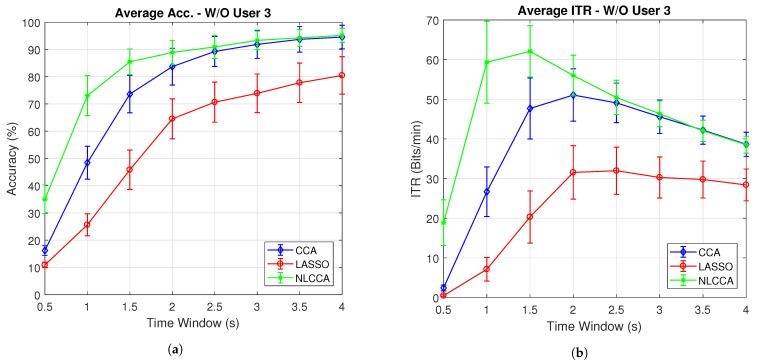
Comparison of (**a**) average accuracy and (**b**) average ITR (excluding User 3) between CCA, LASSO, and NLCCA.

**Figure 9 sensors-21-05308-f009:**
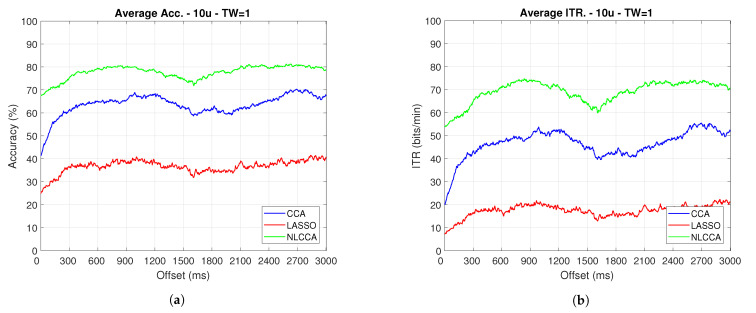
One-second sliding window for (**a**) Accuracy and (**b**) ITR.

**Figure 10 sensors-21-05308-f010:**
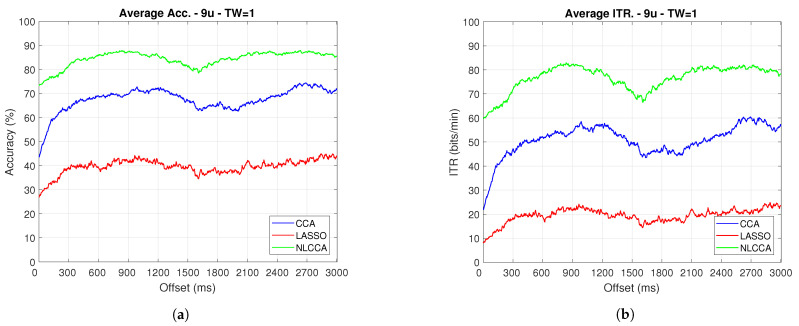
One-second sliding window for (**a**) Accuracy and (**b**) ITR removing User 3.

**Figure 11 sensors-21-05308-f011:**
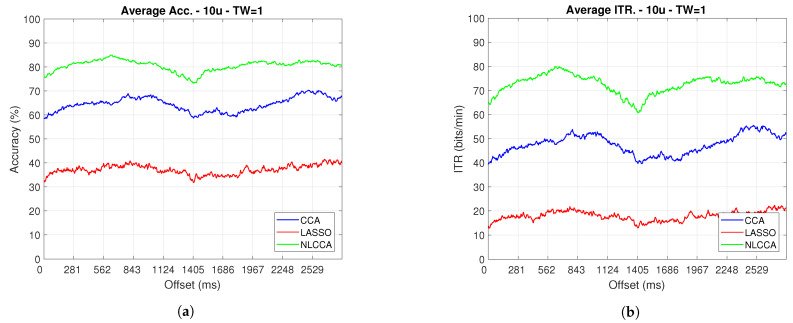
One–second sliding window (trimmed 50 first data—200 ms) of (**a**) Accuracy and (**b**) ITR.

**Figure 12 sensors-21-05308-f012:**
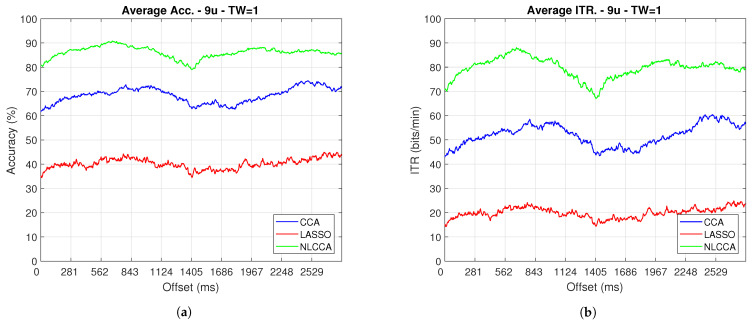
One–second sliding window (trimmed 50 first data–around 200 ms) of (**a**) Accuracy and (**b**) ITR removing User 3.

**Table 1 sensors-21-05308-t001:** Maximum ITR value for each user related to each method, and the corresponding accuracy and time window values.

	LASSO			CCA			NLCCA		
Subject	TW	Acc	ITR	TW	Acc	ITR	TW	Acc	ITR
	(s)	(%)	(bit/min)	(s)	(%)	(bit/min)	(s)	(%)	(bit/min)
S1	3	34.17	5.72	4	60	14.76	1.5	57.5	27.14
S2	4	79.17	25.51	2.5	88.33	45.63	1	72.5	53.55
S3	0.5	5	0.48	4	77.5	24.45	3.5	25	2.38
S4	2.5	71.67	29.91	3	93.33	45.01	2	81.67	45.27
S5	3	90.83	42.39	2	97.5	66.6	1	90	83.1
S6	4	68.33	19.07	2	92.5	58.82	1	95.83	95.73
S7	2	81.67	45.27	2	98.33	68.1	1	86.67	76.72
S8	2	83.33	47.17	2	90	55.4	1.5	81.67	54.32
S9	2	99.17	69.73	1.5	99.167	83.68	1	95.83	95.73
S10	3.5	71.67	23.26	2.5	93.33	51.45	1.5	90	66.48
**Avg.**	**2.65**	**68.50**	**30.85**	**2.55**	**88.99**	**51.39**	**1.5**	**77.67**	**60.04**
**Avg. (w/o S3)**	**2.89**	**75.56**	**34.23**	**2.39**	**90.28**	**54.38**	**1.28**	**83.52**	**66.45**

**Table 2 sensors-21-05308-t002:** Mean and standard deviation of the offset plot.

		LASSO			CCA			NLCCA		
		**Mean**	**STD**	**CV (%)**	**Mean**	**STD**	**CV (%)**	**Mean**	**STD**	**CV (%)**
**Accuracy**	**10 user**	37.20	1.94	5.21	64.44	3.00	4.66	80.54	2.37	2.95
	**9 user**	40.19	2.12	5.28	68.41	3.15	4.61	86.14	2.37	2.75
**ITR**	**10 user**	17.95	1.91	10.64	47.54	4.16	8.74	72.58	3.94	5.43
	**9 user**	19.91	2.12	10.65	51.95	4.50	8.67	79.65	4.22	5.30

**Table 3 sensors-21-05308-t003:** Analysis of Variance (ANOVA).

Case	Source Variability	GDL	Sum Squares	Mean Squares	Fc	F (95%)
**Accuracy (10 users)**	Treatments	2	690,084.705	345,042.35	56,214.05	3
	Error	2154	13,221.27	6.138		
	Total	2156	703,305.98			
	**56,214.05 > 3**					
**Accuracy (9 users)**	Treatments	2	772,312.22	386,156.11	57,822.38	3
	Error	2154	14,385.1	6.68		
	Total	2156	786,697.31			
	**57,822.38 > 3**					
**ITR (10 users)**	Treatments	2	1,075,514.163	537,757.08	44,268.26	3
	Error	2154	26,166.12	12.15		
	Total	2156	1,101,680.28			
	**44,268.26 > 3**					
**ITR (9 users)**	Treatments	2	1,285,243.27	642,621.635	45,287.186	3
	Error	2154	30,565.09	14.19		
	Total	2156	1,315,808.36			
	**45,287.186 > 3**					

**Table 4 sensors-21-05308-t004:** Post-Tukey’s test for Accuracy performance.

Case	Hypothesis	$\left|X_{i}-X_{j}\right|$	Coefficient	${\mathrm{DVS}}_{c}$	Comparison
			q (5%) _3,2154_	$q\sqrt{\frac{{\mathrm{CM}}_{d}}{n}}$	
**Accuracy (10 users)**	$H_{0}$: $U_{1}=U_{2}$	27.24	3.32	0.306	27.24 > 0.306
	$H_{0}$: $U_{1}=U_{3}$	43.34			43.34 > 0.306
	$H_{0}$: $U_{2}=U_{3}$	16.1			16.1 > 0.306
**Accuracy (9 users)**	$H_{0}$: $U_{1}=U_{2}$	28.22	3.32	0.32	28.22 > 0.32
	$H_{0}$: $U_{1}=U_{3}$	45.95			45.95 > 0.32
	$H_{0}$: $U_{2}=U_{3}$	17.73			17.73 > 0.32
**ITR (10 users)**	$H_{0}$: $U_{1}=U_{2}$	29.59	3.32	0.431	29.59 > 0.431
	$H_{0}$: $U_{1}=U_{3}$	54.63			54.63 > 0.431
	$H_{0}$: $U_{2}=U_{3}$	25.04			25.04 > 0.431
**ITR (9 users)**	$H_{0}$: $U_{1}=U_{2}$	32.04	3.32	0.466	32.04 > 0.466
	$H_{0}$: $U_{1}=U_{3}$	59.74			59.74 > 0.466
	$H_{0}$: $U_{2}=U_{3}$	27.7			27.7 > 0.466

The mean difference is greater than the DVSc value in all cases. Therefore, all null hypotheses are rejected.

## Data Availability

Not applicable.

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
