# Peer review of "Solving the SSVEP Paradigm Using the Nonlinear Canonical Correlation Analysis Approach"

_sensors, 2021, doi:10.3390/s21165308_

Round 1

Reviewer 1 Report

Introduction: very little valuable literature positions cited, and mostly outdated work.

Line 125: why did you choose these particular filters?

Why did you decide to compare your approach with very outdated optics from 2001 and 2008? Nothing new came in the meantime?

The Results section does not put emphasis on the obtained results in a strong way.

Conclusion section states some important facts about the EEG data, but no references given. Some of these information should be added to the introduction.

Bibliography should be updated with more positions from 2020 no 2021.

Paper needs to be proofread.

Author Response

First, we would like to thank you for reviewing our manuscript and for the importance of your comments. Below, you will find point-by-point responses and modifications we have made to address the issues mentioned. We hope that we have given a satisfactory response to all of them and that the paper now reaches the quality required for publication in the MDPI Sensors. Significant changes were highlighted in red type.

Reviewer 2 Report

In your paper, you use the NLCCA model to solve
the SSVEP paradigm. However, some issues must be
resolved.
1. In subsection 2.4.3, Many of the symbolic
representations of Equation (8) are wrong.
2. From Line 189 to Line 193, many comments on
Equation (8) are wrong. For example, In the
formula (8), P_1, but in comments, is P1. Obviously
these two statements don't agree.
3. All three of your formulas(the formulas (9),
(10),(12)) have serious problems and must be
expressed correctly.
4. Since many of your previous formulae are wrong,
your mathematical model of the NLCCA must work
hard to derive it completely correctly. Otherwise, the
authenticity of your subsequent verification cannot be
guaranteed. In addition, your NLCCA mathematical
model should not only derive correctly, but also
explain the essential differences and advantages
between your model and CCA model.
5. All your other formulas must also be derived
correctly, including the writing of symbols.
6. At the conclusion of your fourth section, the
previous part suggests porting it to the third section
because it feels too long.
To sum up, your core mathematical models and
formulas must be derived correctly.

Author Response

(The authors gave the same response as above.)

Round 2

Reviewer 1 Report

Dear authors - I am satisfied with the amendments done.

Author Response

We appreciate very much your feedback and your comments enhanced our article.

Thank you very much

Reviewer 2 Report

Major revised.

Author Response

First of all, we would like to thank you again for reviewing our manuscript and for the importance of your comments. A complete reworking of the wording and language has been carried out. Below you will find a point-by-point response, as well as the modifications we have made to address the issues mentioned, we hope we have provided a satisfactory response to all of them, and that the article now reaches the quality required for publication in MDPI Sensors. Significant changes are highlighted in red type.

Round 3

Reviewer 2 Report

To sum up, there are still small details to be modified. Your paper needs a minor revision.

Author Response

We would like to thank you again for reviewing our manuscript and for the importance of your comments. Below you will find a point-by-point response, as well as the modifications we have made to address the issues mentioned, we hope we have provided a satisfactory response to all of them, and that the article now reaches the quality required for publication in MDPI Sensors. Significant changes are highlighted in red type:
